# Immunogenicity and Durability of Antibody Responses to Homologous and Heterologous Vaccinations with BNT162b2 and ChAdOx1 Vaccines for COVID-19

**DOI:** 10.3390/vaccines10111864

**Published:** 2022-11-04

**Authors:** Dong-In Kim, Seo Jin Lee, Soonju Park, Paul Kim, Sun Min Lee, Nakyung Lee, David Shum, Dong Ho Kim, Eui Ho Kim

**Affiliations:** 1Viral Immunology Laboratory, Institut Pasteur Korea, Seongnam 13488, Korea; 2Department of Pediatrics, Korea Cancer Center Hospital, Seoul 01812, Korea; 3Screening Discovery Platform, Institut Pasteur Korea, Seongnam 13488, Korea

**Keywords:** COVID-19, SARS-CoV-2, vaccine, BNT162b2, ChAdOx1, antibody

## Abstract

During the COVID-19 pandemic, vaccines were developed based on various platform technologies and were approved for emergency use. However, the comparative analysis of immunogenicity and durability of vaccine-induced antibody responses depending on vaccine platforms or vaccination regimens has not been thoroughly examined for mRNA- or viral vector-based vaccines. In this study, we assessed spike-binding IgG levels and neutralizing capacity in 66 vaccinated individuals prime-boost immunized either by homologous (BNT162b2-BNT162b2 or ChAdOx1-ChAdOx1) or heterologous (ChAdOx1-BNT162b2) vaccination for six months after the first vaccination. Despite the discrepancy in intervals for the prime-boost vaccination regimen of different COVID-19 vaccines, we found stronger induction and relatively rapid waning of antibody responses by homologous vaccination of the mRNA vaccine, while weaker boost effect and stable maintenance of humoral immune responses were observed in the viral vector vaccine group over 6 months. Heterologous vaccination with ChAdOx1 and BNT162b2 resulted in an effective boost effect with the highest remaining antibody responses at six months post-primary vaccination.

## 1. Introduction

COVID-19, a disease caused by SARS-CoV-2, has been a major health threat that is spreading rapidly worldwide. SARS-CoV-2 infection was first reported in Wuhan, China, in November 2019, and the WHO declared it a pandemic on 11 March 2020 [1]. Vaccine development progressed rapidly, and an mRNA vaccine BNT162b2 (BNT) and an adenoviral vector vaccine ChAdOx1 (ChAd) were the first vaccines in each vaccine platform approved for emergency use at the end of the same year [2,3,4]. At that time, the viral vector vaccine was a recently added technology, and the mRNA vaccine had not been licensed. Both COVID-19 vaccines were reported to have strong protective efficacies (BNT: 95%; ChAd: 70.4%) with potent humoral immune responses to the original SARS-CoV-2 strain and decent levels of T-cell responses [3,4]. It has been demonstrated that protective efficacy is closely associated with the levels of spike-binding and neutralizing antibodies and suggested as potential immune correlates of protection [5,6]. A vast number of studies have characterized COVID-19 vaccine-induced immune responses in human vaccinees and revealed stronger induction of spike-binding and neutralizing antibodies by the BNT vaccine than the ChAd vaccine. In addition, enhanced protective immunity by heterologous ChAd-BNT vaccinations was reported [7,8,9,10,11,12,13,14,15,16,17,18,19]. However, there is a paucity of knowledge describing the immunogenicity and long-term maintenance of vaccine-induced immunity comparing homologous and heterologous vaccinations.

In South Korea, ChAd and BNT vaccines were introduced and utilized for mass vaccination programs during the early period. The dominant interval for the prime-boost vaccination regimen for the ChAd vaccine was 10–12 weeks, while that for the BNT vaccine was three weeks. Although most of the two-shot immunizations were carried out in a homologous manner, the BNT vaccine was also used for secondary vaccination of some ChAd vaccinees. In this study, we followed vaccinated individuals in South Korea for up to six months after the first vaccination to examine the immunogenicity and longevity of antibody responses to homologous and heterologous vaccinations of BNT and ChAd vaccines.

## 2. Materials and Methods

### 2.1. Study Design

Vaccination programs against COVID-19 are being conducted nationwide in South Korea. Our study included subjects uninfected with SARS-CoV-2 and vaccinated with either the BNT or ChAd vaccine (Table 1). Subsequently, whole blood specimens were collected from those vaccinated with the BNT vaccine 3–4 weeks after the first vaccination and three weeks, three months, and 5–6 months after the second vaccination. From the ChAd-vaccinated subjects, blood samples were gathered 3–4 weeks after the first vaccination. For the second vaccination, either the ChAd vaccine or the BNT vaccine was administered. Blood samples were collected at four weeks and three months after the second vaccination.

### 2.2. Separation of Plasma Specimen

Whole blood specimens were collected, and plasma and peripheral blood mononuclear cells (PBMCs) were isolated using cell preparation tubes (CPT) vacutainer (BD Biosciences). The CPT Vacutainer tubes were centrifuged at 1800× *g* for 20 min at 4 °C, resulting in the separation of blood into plasma and PBMCs. After the centrifugation, plasma corresponding to the supernatant was carefully harvested.

### 2.3. Cells and Virus

Vero cells were obtained from the American Type Culture Collection (ATCC, CCL-81) and cultured in Dulbecco’s modified Eagle medium (DMEM; Welgene, Korea) and maintained in a humidified environment at 37 °C in the presence of 5% CO_2_. Cell culture medium contained 1% penicillin/streptomycin (PS; Welgene, Korea) and 10% heat-inactivated fetal bovine serum (FBS; GIBCO). SARS-CoV-2 (βCoV/Korea/KCDC/2020; NCCP43326) isolated in February 2020 was provided by the Korea Disease Control and Prevention Agency (KDCA). Viral amplification and titration were performed using Vero cells.

### 2.4. Proteins and Antibodies

The SARS-CoV-2 spike protein and nucleocapsid (NP) protein were purchased from Sino Biologicals (40589-V08B1 and 40588-V08B). Antibodies, including goat anti-human IgG (H+L)-UNLB, human IgG-UNBL, and horseradish peroxidase-conjugated goat anti-human IgG, were purchased from SouthernBiotech.

### 2.5. Enzyme-Linked Immunosorbent Assay (ELISA)

Briefly, 96-well high-binding EIA/RIA plates (Costar) were coated with 50 μL/well of 1 ug/mL spike or NP proteins in phosphate-buffered saline (PBS) overnight. The coated plates were washed twice with 200 μL/well of PBS containing Tween 20 (PBS-T), blocked with blocking buffer (1% blotting-grade blocker (BIO-RAD) in PBS-T) for 30 min at 37 °C. Plasma samples were serially diluted 4-folds in blocking buffer, added to the plates, and incubated at room temperature (RT) for 2 h. The plates were washed three times with PBS-T, and HRP-conjugated goat anti-human IgG (SouthernBiotech) in blocking buffer was added, followed by incubation for 1.5 h at RT. After washing three times with PBS-T, the wells were treated with TMB substrate solution (OptEIA reagent set, BD). Finally, the reaction was stopped by the addition of a stop solution (0.5 M hydrochloric acid). Optical density at 450 nm was measured using a microplate reader (Victor 3, PerkinElmer).

### 2.6. Microneutralization (MN) Assay

Vero cells were seeded at 15,000 cells per well in Opti-PRO™ SFM (Gibco) with 1X antibiotic-antimycotic solution and 4 mM L-glutamine (Gibco) in a 96-well clear plate (Greiner Bio-One) 24 h prior to the experiment. Plasma samples were diluted 3-fold in quadruplicates and mixed with an equal volume of 100 tissue culture infective dose 50% (100 TCID50) SARS-CoV-2 virus. After pre-incubation for 30 min at 37 °C, plasma/virus mixtures were transferred to Vero cells. After 96 h, the cytopathic effect of SARS-CoV-2 on the infected cells was measured via bright-field imaging. The neutralizing antibody (nAb) titer was calculated as the reciprocal of the highest test plasma dilution factor at which 50% neutralization was attained.

### 2.7. Statistical Analyses

Error bars indicate the standard error of the mean (SEM). *p* values were analyzed using the Mann–Whitney U test for two-group data and the Kruskal–Wallis test for three-group data. The analysis was conducted using Prism 8 (GraphPad Prism Software, San Diego, CA, USA).

## 3. Results

### 3.1. Study Design and Participants

Overall, 66 participants were enrolled in this study and were vaccinated twice with COVID-19 vaccines, including BNT162b (BNT, mRNA vaccine) and ChAdOx1 (ChAd, viral-vector vaccine), in three different ways: (1) BNT162b × BNT162b (BNT-BNT), (2) ChAdOx1 × ChAdOx1 (ChAd-ChAd), and (3) ChAdOx1 × BNT162b (ChAd-BNT) (Figure 1A and Table 1). In total, 46 participants received homologous two-time vaccination with BNT162b at a 3-week interval. The remaining 22 participants received a ChAdOx1 immunization as the primary vaccine. Approximately three months later, among the 22 ChAdOx1 vaccines, nine participants received the second shot with the ChAdOx1 vaccine, while thirteen participants received the BNT162b vaccination, representing the heterologous prime-boost vaccination. The majority (60–88.9%) of the participants in all three groups were women, but there was no statistical difference in sex distribution among the groups. The median age of the participants in each group was 41, 53, and 40 years for the BNT-BNT, ChAd-ChAd, and ChAd-BNT groups, respectively. All participants were healthy and had no history of SARS-CoV-2 infection or serious underlying diseases at the time of vaccination (Figure 1B). To analyze antibody responses, blood specimens were collected approximately at 3–4 weeks post-primary post-vaccination in all three groups (Figure 1A). In the BNT-BNT group, longitudinal blood specimens were collected at approximately three weeks, 14 weeks, and 5–6 months after the second vaccination. In the ChAd-ChAd and ChAd-BNT groups, additional blood specimens were collected approximately three weeks and three months after the second vaccination. The last time point for blood collection was approximately 5.5–6.5 months after the first vaccination.

### 3.2. Induction of Antibody Responses after the First and Second Vaccinations

Antibody responses, including spike-binding and neutralizing antibodies (nAbs), have been suggested as key protective indicators of SARS-CoV-2 infection [5,6]. First, we measured the spike-specific immunoglobulin G (IgG) levels in plasma specimens 3–4 weeks after the first vaccination. As a primary vaccine-induced immune response, BNT vaccination resulted in a significantly higher spike-specific IgG level than ChAd vaccination (Figure 1C). In addition, the neutralizing capacity was slightly stronger in the BNT group than in the ChAd group after the primary vaccination (Figure 1D). After the second vaccination, the BNT-BNT group triggered a potent boost effect and generated a higher level of spike-binding antibodies than the ChAd-ChAd group (Figure 1E). Interestingly, the ChAd-BNT group displayed IgG levels comparable to those of the BNT-BNT-vaccinated people. Levels of SARS-CoV-2 nAbs showed a similar trend to that of the binding antibodies (Figure 1F). Overall, BNT-BNT homologous vaccination and ChAd-BNT heterologous vaccination resulted in stronger antibody responses than ChAd-ChAd homologous vaccination.

### 3.3. Longitudinal Analysis of Spike-Binding Antibody and Neutralizing Antibody Responses

Next, to estimate the durability of antibody responses, we tracked the kinetic changes in spike-binding antibodies from the three vaccination groups until approximately six months after the first vaccination. After the peak of the spike-specific antibody response at 3 weeks post-boost, the antibody level started to decay over time in the BNT-BNT group (Figure 2A). In the ChAd-ChAd group, the second vaccination did not substantially enhance the binding antibody response, although the interval between the two vaccinations was much longer (three months) than that in the BNT-BNT group (three weeks). Intriguingly, in the ChAd-ChAd-vaccinated group, spike-specific antibody levels remained stable for the next three months (Figure 2B). The ChAd-BNT group displayed a significantly augmented secondary antibody response, followed by a moderate decrease three months after the second shot (Figure 2C). As a result, at approximately six months after the first vaccination (the last time point), in contrast to the peak antibody responses, the heterologous vaccination maintained slightly higher levels of antibodies than the BNT-BNT groups and similar antibody levels in the BNT-BNT and ChAd-ChAd groups (Figure 2D).

**Figure 1 vaccines-10-01864-f001:**
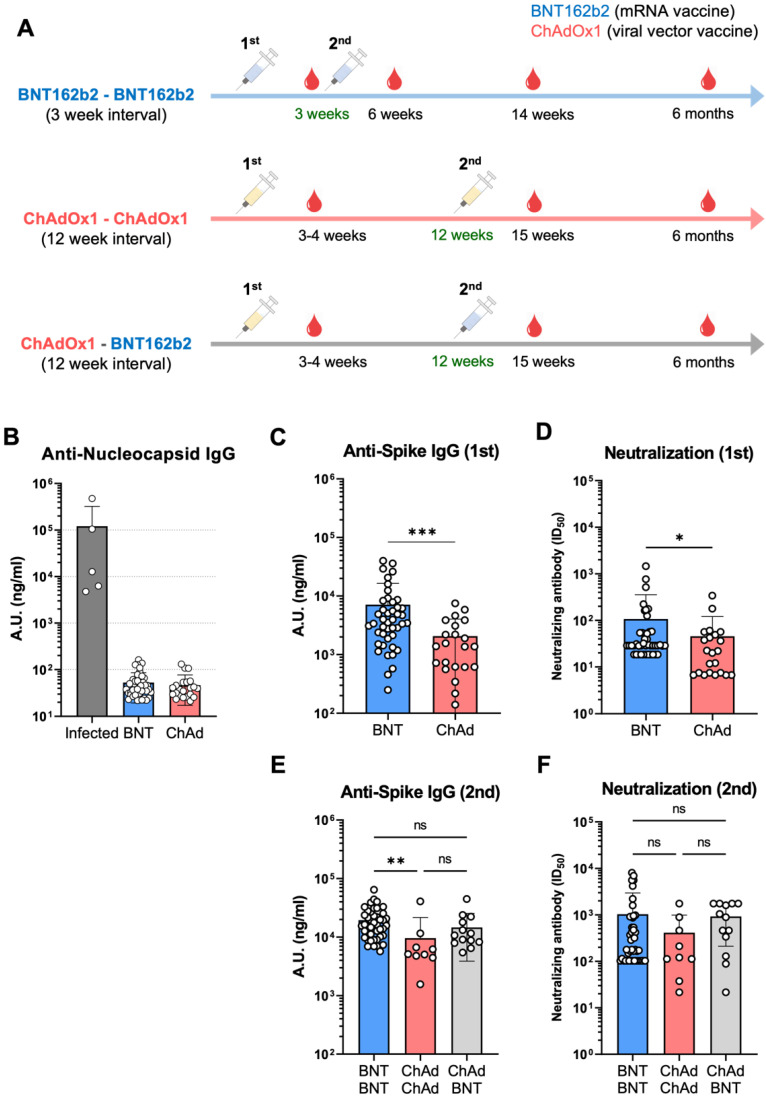
Study design and peak antibody responses. (**A**) Schematic presentation of experimental schedule. The individuals were vaccinated with BNT162b2 and ChAdOx1 followed by the second dose in either homologous (BNT162b2-BNT162b2, ChAdOx1-ChAdOx1) or heterologous (ChAdOx1-BNT162b2) manner. The groups and time points for collecting blood specimens are depicted in the figure. (**B**) Levels of anti-nucleocapsid antibody in plasma specimens. (**C**,**D**) Levels of SARS-CoV-2 spike-binding antibody (**C**) and neutralizing antibody to SARS-CoV-2 (**D**) were measured from plasma specimens at 3–4 weeks post-primary vaccination by ELISA and microneutralization assay, respectively. (**E**,**F**) Titers of SARS-CoV-2 spike-binding antibody (**E**) and neutralizing antibody (**F**) were assessed three weeks after the boost shot. * *p* < 0.05, ** *p* < 0.01, and *** *p* < 0.001.

Based on these kinetic changes, we analyzed the fold differences in spike-specific antibody responses at different time points (Figure 2E–G). Despite the overall fold-increase in spike-binding antibodies after the second vaccination in all three vaccinated groups, the heterologous vaccination (ChAd-BNT) group displayed the strongest boost effect (Figure 2E). Since the interval between the first and second shots vary among different vaccination groups, we tried to analyze the durability of binding antibody responses in this aspect. As shown in Figure 2F, the fold changes for 3 months after the second vaccination were calculated, suggesting a more rapid decrease in IgG levels in the BNT-BNT group but a slightly better persistence in the ChAd-ChAd group. We also assessed the durability of spike-binding antibodies by comparing the antibody levels at three weeks and six months after the first vaccination (Figure 2G). Remarkably, heterologous vaccination (ChAd-BNT) resulted in a 4.4-fold increase in the level of spike-specific antibodies. The BNT-BNT group showed slightly decreased spike-binding antibodies, while the ChAd-ChAd group remained about the same.

In addition to the spike-binding antibodies, we examined the kinetic changes in nAbs. In all three groups, the second vaccination induced an increase in nAbs (Figure 3A–C), resulting in a similar trend for spike-binding antibodies, although there was no statistical difference (Figure 3D). The magnitude of nAb increase after the second vaccination was the highest in the ChAd-BNT group (82.9-fold), followed by the BNT-BNT group (35-fold) (Figure 3E). After the secondary peak IgG responses, both the BNT-BNT and ChAd-BNT groups resulted in the sharp waning of nAb titers, whereas the ChAd-ChAd group could stably maintain nAbs (Figure 3F). Over the course of the experiment, heterologous vaccination with ChAd-BNT efficiently triggered the accumulation of nAbs against SARS-CoV-2, and either the BNT-BNT or ChAd-ChAd groups displayed a moderate increase in nAbs (Figure 3G). Taken together, these data demonstrate the unique dynamics of vaccine-induced humoral immune responses depending on different vaccine platforms and vaccination regimens, particularly indicating the potent immunogenicity and long-lasting antibody response of the heterologous ChAd-BNT vaccination.

## 4. Discussion

Recently, numerous studies have investigated humoral and cellular immune responses triggered by various COVID-19 vaccines and their protective capacity against diverse variants [20,21,22,23]. Evidence on the durability of vaccine-induced immunity depending on vaccine platform technologies and administration regimens is essential because it can impact public health policies, such as vaccine choice and timing of booster vaccination. In this study, we addressed the induction and longevity of antibody responses by homologous or heterologous vaccinations of the mRNA vaccine BNT162b2 and the viral-vector vaccine ChAdOx1. Here, despite relatively small cohort sizes for the ChAd-ChAd and ChAd-BNT groups (Table 1), we revealed that the BNT-BNT group showed strong formation and a relatively rapid decrease in humoral immune response, whereas the ChAd-ChAd vaccination generated moderate antibody responses with stable maintenance. Strikingly, the BNT boost after ChAd priming elicited a very potent boost effect, resulting in the highest levels of binding and neutralizing antibodies 6 months after the first vaccination (Figure 2C and Figure 3C).

First, the different working mechanisms of mRNA vaccines and viral-vector vaccines may influence the modalities of vaccine-induced immune responses. Several previous reports have described superior induction of humoral immune responses by mRNA vaccine platforms, including BNT162b2 and mRNA-1273, in contrast to the relatively stable durability of the viral-vector vaccine [7,9,24,25,26]. We observed a clear difference in the kinetic changes in the antibody responses between the ChAd-ChAd and ChAd-BNT groups. After ChAd vaccination, the BNT boost effectively increased the levels of binding and neutralizing antibodies despite a mild increase in the ChAd boost vaccination (Figure 2B,C and Figure 3B,C). It is possible that the generation of viral vector-specific antibodies upon primary ChAd vaccination inhibited efficient delivery, resulting in attenuated subsequent vaccine-induced immunity.

Moreover, in addition to the discrepancy in vaccine platforms, the interval between the first and second shots may be another factor affecting the longevity of the immune response. According to previous studies, sufficient time is required to establish good-quality immunological memory [27,28,29]. Our data from the BNT-BNT group may imply that a longer than 3-week interval is optimal for steady maintenance of the antibody response (Figure 2G and Figure 3G).

In our study, the neutralizing capacity of the variant strains of SARS-CoV-2 was not measured because the focus of our study was to investigate the immunogenicity and longevity of the antibody response to the original vaccine antigen. It is noteworthy that the durability of humoral response is primarily determined by memory B cells and long-lived plasma cells, and it was shown that the spike-specific memory B cells persisted for a longer time, although the levels of binding antibodies declined over 8 months [30]. We did not assess vaccine-induced T-cell responses that could support the germinal center reaction or T-dependent antibody response. Examining the immunogenicity and durability of T-cell responses depending on different vaccine platforms will provide a beneficial contribution to the vaccine field.

In conclusion, we identified distinct immunogenicity and durability of vaccine-induced immune responses, depending on various vaccine development technologies and vaccination strategies. Homologous vaccination with BNT and ChAd vaccines displayed strong induction and the rapid decay of antibody response and moderate induction and slow waning of antibodies, respectively. Heterologous vaccination with ChAd and BNT triggered a potent boosted antibody response, and the response remained at the highest level at approximately 6 months after the primary vaccination. The data obtained in this study will be advantageous in determining an effective vaccine strategy for public health.

## Figures and Tables

**Figure 2 vaccines-10-01864-f002:**
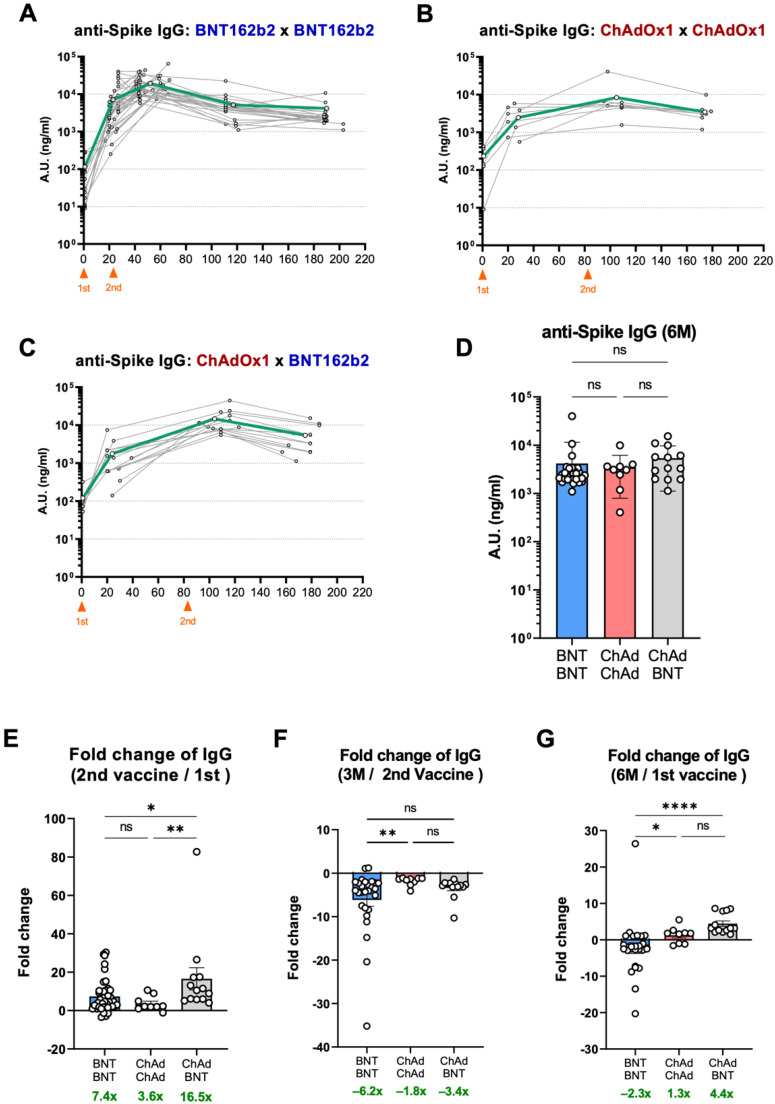
Kinetic changes and maintenance of spike-binding IgGs. Vaccine-induced spike-binding antibody titers were longitudinally monitored in three vaccinated groups. (**A**–**C**) Kinetics of SARS-CoV-2 spike-specific IgG levels in the BNT-BNT (**A**), ChAd-ChAd (**B**), and ChAd-BNT (**C**) groups. (**D**) Titers of anti-spike IgGs in the three groups approximately six months after primary immunization. (**E**) Boosting effect of secondary vaccination depicted by the fold increase in anti-spike IgG levels. (**F**) Fold change in anti-spike IgG contractions for three months after the second shot. (**G**) Fold changes in anti-spike IgG titers from three weeks post-primary immunization to the last monitoring period (six months). * *p* < 0.05, ** *p* < 0.01, and **** *p* < 0.0001.

**Figure 3 vaccines-10-01864-f003:**
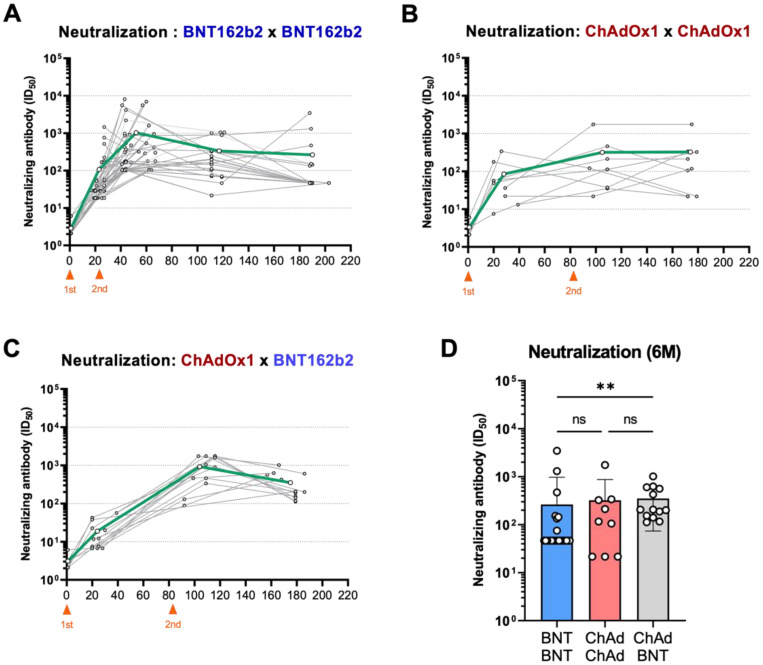
Kinetic changes and maintenance of SARS-CoV-2 neutralizing antibodies. Plasma-neutralizing antibody titers against SARS-CoV-2 were assessed longitudinally in vaccinated individuals. (**A**–**C**) Kinetic changes in neutralizing antibody titers in the BNT-BNT (**A**), ChAd-ChAd (**B**), and ChAd-BNT (**C**) groups. (**D**) Neutralizing capacities in the three groups of vaccinees at approximately six months after primary immunization. (**E**) Boosting effect of secondary vaccination is displayed by the fold increase in neutralizing antibody levels. (**F**) Fold change in neutralizing antibodies for 3 months after the second shot. (**G**) Fold changes in neutralizing antibody titers from three weeks to approximately six months after the primary immunization. ** *p* < 0.01, and *** *p* < 0.001.

**Table 1 vaccines-10-01864-t001:** Summary of participants.

Total (N = 66)		BNT162b2—BNT162b2 (N = 44)			ChAdOx1—ChAdOx1 (N = 9)	ChAdOx1—BNT162b2 (N = 13)
		Short-Term (N = 15)	Long-Term (N = 29)	Total (N = 44)		
**Sex**	**Female**	9 (60%)	24 (82.8%)	34 (73.9%)	8 (88.9%)	9 (69.2%)
	**Male**	6 (40%)	5 (17.2%)	12 (26.1%)	1 (11.1%)	4 (30.8%)
**Median age** (Range)		44 (18–58)	38 (24–56)	41 (18–58)	53 (48–58)	40 (32–48)
**1st vaccine response** Days after the 1st shot		17–26	20–27	17–27	20–39	20–39
**2nd vaccine response** Days after the 1st shot (Days after the 2nd shot)		47–70 (16–36)	41–44 (21–24)	41–70 (16–36)	95–109 (19–28)	92–123 (20–32)
**Long-term time points** Days after the 1st shot (Days after the 2nd shot)			111–121 (91–101); 188–203 (168–186)		172–179 (91–98)	157–186 (84–103)

## Data Availability

Not applicable.

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
