# Peer review of "Immunogenicity and Durability of Antibody Responses to Homologous and Heterologous Vaccinations with BNT162b2 and ChAdOx1 Vaccines for COVID-19"

_vaccines, 2022, doi:10.3390/vaccines10111864_

Round 1

Reviewer 1 Report

Kim et al. investigate the antibody responses directed against the ancestral SARS-CoV-2 Spike in cohorts of vaccines which received two doses (homologous) of Pfizer/BioNTech (BNT162b2) or AstraZeneca/Oxford (ChAdOx1) COVID-19 vaccines, or one dose of each vaccine (heterologous). Individuals were followed up for up to 6 months after the first vaccine dose to observe the durability of the response. While these results are interesting and in line with the literature, this study is more of a confirmatory study rather than providing novel results.

 Major comments:

1. In contrast with the authors’ statement, the immunogenicity and durability of COVID-19 mRNA-based vaccines and adenoviral vector vaccines, in homologous or heterologous vaccination settings, have been extensively described in the literature. It is problematic that most of these studies are only acknowledge in the discussion, while the abstract and introduction present the study has showing completely novel findings. The literature on the subject should be properly acknowledge in the introduction and the novelty claims should be removed (lines 17-19 and 51-54). Additional studies should also be cited (PMID: 33991480, 34181880, 34260850, 34262158, 34312554, 34332707, 34451982, 34452043, 34929493, 35057798, 35203547, 35334989, 35455240, 35580156, 35632405, 35637788, 35764089, 35799790, 35946811, 35953492, 36111560).

2. Antibody titers are not the best predictors to quantify the long-lasting humoral immunity against SARS-CoV-2. Multiple studies have shown that memory B cells specific for SARS-CoV-2 Spike persist for months after vaccination, even after antibody titers have declined. This should discussed.

3. Figure 1A: The interval between vaccine doses has been previously shown by multiple groups to modulate the antibody responses against SARS-CoV-2 Spike. The difference in interval between BNT-BNT and ChAd-BNT groups makes it impossible to conclude which vaccine regimen is superior. The abstract should be modified accordingly.

4. Table 1: The cohort size is relatively small, especially for the ChAd group (10 individuals) and the ChAd/BNT group (13 individuals). This should be emphasized in the discussion.

5. Lines 38-40: Authors state that BNT162b2 and ChAdOx1 vaccines were the first COVID-19 vaccines to be approved for emergency use in the end of 2020. This is not fully true since the Moderna vaccine (mRNA-1273) got the approval before the ChAdOx1 vaccine.

6. Line 69-70 and 153: Individuals enrolled in this protocol are described as being “uninfected” and having “no history of SARS-CoV-2 infection”. How was this controlled? Authors should provide anti-nucleocapsid antibody titers to confirm this statement and rule out the possibility of previous asymptomatic SARS-CoV-2 infection since it could represent an important confounding factor in the quantification of anti-Spike antibody titers for this cohort.

7. Lines 93, 315, 335: The SARS-CoV-2 strain used for microneutralization assay is said to come from a 2020 Korean isolate, but authors mentioned it as a “Wuhan strain”. Since this strain does not come from Wuhan but Korea, it should rather be labelled with the name of its lineage using the WHO, Pango or Nextstrain nomenclature to avoid any confusion.

8. Lines 131-132: There is some concern about the statistical analysis. Authors used exclusively Student T test to perform all their statistical analyses, without controlling for data distribution normality. Especially in the context of this small cohort, data distribution that are non-parametric should be analyzed with Mann-Whitney U test instead of Student t test. This could explain some odd statistical significance like in Figure 2D.

9. The y axis of Figure 2D is cut at 10e2 while all previous data (Figure 2A-C) have the y axis going down to 10e0. This should be corrected to be uniform.

10. For Figure 2A-C and Figure 3A-C, it would useful to add a thicker line depicting the average/median longitudinal values for the vaccination group.

11. For Figure 2E-G and Figure 3E-G, fold changes for each individual should be shown as a dot plot rather than an average histogram. Therefore, statistical analysis should be performed to see if the the differences in fold changes are significant.

Author Response

* Reviewer 1

Kim et al. investigate the antibody responses directed against the ancestral SARS-CoV-2 Spike in cohorts of vaccines which received two doses (homologous) of Pfizer/BioNTech (BNT162b2) or AstraZeneca/Oxford (ChAdOx1) COVID-19 vaccines, or one dose of each vaccine (heterologous). Individuals were followed up for up to 6 months after the first vaccine dose to observe the durability of the response. While these results are interesting and in line with the literature, this study is more of a confirmatory study rather than providing novel results.

Major comments:

  1. In contrast with the authors’ statement, the immunogenicity and durability of COVID-19 mRNA-based vaccines and adenoviral vector vaccines, in homologous or heterologous vaccination settings, have been extensively described in the literature. It is problematic that most of these studies are only acknowledge in the discussion, while the abstract and introduction present the study has showing completely novel findings. The literature on the subject should be properly acknowledge in the introduction and the novelty claims should be removed (lines 17-19 and 51-54). Additional studies should also be cited (PMID: 33991480, 34181880, 34260850, 34262158, 34312554, 34332707, 34451982, 34452043, 34929493, 35057798, 35203547, 35334989, 35455240, 35580156, 35632405, 35637788, 35764089, 35799790, 35946811, 35953492, 36111560).

* Response: The authors appreciate the reviewer’s valuable suggestion. As the reviewer indicated, those important publications demonstrate either immunogenicity or longevity in homologous or heterologous vaccinations. We tried to add these seminal publications in the introduction section as many as possible. In addition, the authors wanted to describe the immunogenicity and longevity of homologous/heterologous COVID-19 vaccinations simultaneously in this study, so the authors believe that this manuscript would provide valuable information to the field.

  1. Antibody titers are not the best predictors to quantify the long-lasting humoral immunity against SARS-CoV-2. Multiple studies have shown that memory B cells specific for SARS-CoV-2 Spike persist for months after vaccination, even after antibody titers have declined. This should discussed.

* Response: The authors thank for the important point. According to the reviewer’s comment, we added the point in the discussion as following.

Line 283: It is noteworthy that the durability of humoral response is primarily determined by memory B cells and long-lived plasma cells, and it was shown that the spike-specific memory B cells persisted for longer time although the levels of binding antibodies declined over 8 months.

  1. Figure 1A: The interval between vaccine doses has been previously shown by multiple groups to modulate the antibody responses against SARS-CoV-2 Spike. The difference in interval between BNT-BNT and ChAd-BNT groups makes it impossible to conclude which vaccine regimen is superior. The abstract should be modified accordingly.

* Response: The authors appreciate the valid point from the reviewer. We tried to extract valuable information as much as possible from the data by comparing ChAd-ChAd and ChAd-BNT groups since these groups have similar interval time.

In addition, as pointed by the reviewer, regarding the comparison between BNT-BNT and ChAd-BNT or ChAd-ChAd groups, it was not feasible to directly compare the kinetics of antibody responses due to the difference in intervals. However, the authors think that it would be valuable to roughly describe the overall changes of humoral responses in three vaccinated groups over 6 months. The abstract was modified accordingly.

  1. Table 1: The cohort size is relatively small, especially for the ChAd group (10 individuals) and the ChAd/BNT group (13 individuals). This should be emphasized in the discussion.

* Response: As suggested by the reviewer, we emphasized this matter by adding a sentence like below.

Line 251: Here, despite relatively small cohort sizes for the ChAd-ChAd and ChAd-BNT groups (Table 1)

  1. Lines 38-40: Authors state that BNT162b2 and ChAdOx1 vaccines were the first COVID-19 vaccines to be approved for emergency use in the end of 2020. This is not fully true since the Moderna vaccine (mRNA-1273) got the approval before the ChAdOx1 vaccine.

* Response: The authors appreciate the valid point from the reviewer. We corrected the sentence like below.

Line 41: Vaccine development progressed rapidly, and a mRNA vaccine BNT162b2 (BNT) and an adenoviral vector vaccine ChAdOx1 (ChAd) were the first vaccines in each vaccine platform, approved for emergency use at the end of the same year.

  1. Line 69-70 and 153: Individuals enrolled in this protocol are described as being “uninfected” and having “no history of SARS-CoV-2 infection”. How was this controlled? Authors should provide anti-nucleocapsid antibody titers to confirm this statement and rule out the possibility of previous asymptomatic SARS-CoV-2 infection since it could represent an important confounding factor in the quantification of anti-Spike antibody titers for this cohort.

* Response: The authors really appreciate the important point from the reviewer. Initially, we had sorted the “uninfected” individuals by the PCR-based SARS-CoV-2 diagnosis kit. However, as suggested by the reviewer, the authors measured the anti-nucleocapsid antibodies in plasma since the PCR-based diagnosis may not able to detect all the previous SARS-CoV-2 infections. Based on the ELISA result, we removed three specimens (two from BNT-BNT group, one from ChAd-ChAd group) displaying high anti-NP IgG titer, re-analyzed all the data and updated the table and figures accordingly.

  1. Lines 93, 315, 335: The SARS-CoV-2 strain used for microneutralization assay is said to come from a 2020 Korean isolate, but authors mentioned it as a “Wuhan strain”. Since this strain does not come from Wuhan but Korea, it should rather be labelled with the name of its lineage using the WHO, Pango or Nextstrain nomenclature to avoid any confusion.

* Response: The authors thank for the valid comment. We deleted “Wuhan strain” and described like following.

Line 98: SARS‐CoV‐2 (βCoV/KOR/KCDC03/2020; NCCP43326) isolated in February 2020 was provided by Korea Centers for Disease Control and Prevention Agency (KDCA).

  1. Lines 131-132: There is some concern about the statistical analysis. Authors used exclusively Student T test to perform all their statistical analyses, without controlling for data distribution normality. Especially in the context of this small cohort, data distribution that are non-parametric should be analyzed with Mann-Whitney U test instead of Student t test. This could explain some odd statistical significance like in Figure 2D.

* Response: The authors appreciate the important point from the reviewer. As suggested, we applied the Mann-Whitney U test for two-group data and the Kruskal-Wallis test for three-group data.

  1. The y axis of Figure 2D is cut at 10e2 while all previous data (Figure 2A-C) have the y axis going down to 10e0. This should be corrected to be uniform.

* Response: The authors thank for the comment. We have changed the scale of the y axis down to 10e0.

  1. For Figure 2A-C and Figure 3A-C, it would useful to add a thicker line depicting the average/median longitudinal values for the vaccination group.

* Response: The authors appreciate wonderful suggestion from the reviewer. Thicker lines depicting average values were added to the kinetic graphs.

  1. For Figure 2E-G and Figure 3E-G, fold changes for each individual should be shown as a dot plot rather than an average histogram. Therefore, statistical analysis should be performed to see if the the differences in fold changes are significant.

* Response: The authors appreciate the important comment from the reviewer. The fold changes values were re-calculated individually, plotted, and statistically tested.

Reviewer 2 Report

The article entitled “Immunogenicity and durability responses to homologous and hererologous vaccinations with BNT162b2 and ChAdOx1 vaccines for COVID-19” reports a study enrroling 69 participants vaccinated with COVId-19 vaccines, including BNT-BNT mRNA homologous vaccination and ChAd-ChAd viral-vector homologous vaccination and ChAd-BNT heretorologous vaccination to evaluate the immunogenicity and longevity of the antobody response to the original vaccine antigen. The authors revealed that the BNT-BNT group and ChAd-ChAd group showed strong induction and rapid decay of antibody response, and moderate induction and slow waning of antibodies, respectively. The ChAd-BNT group showed a potent antibody response, and the response remained at the highest level at approximately  6 months after the primary vaccination. The authors did not assess vaccine-induced T-cell and B-cell responses that in my opinion can provide a better understanding about the different immunogenicity and durability of vaccine-induced immune response.

 Although there are many articles about this argument in the literature, I think that this manuscript reinforces the concept of the usefulness of vaccination practice and this is a strenght point in this manuscript. The weakness is that the authors would provide more clinical and biological details (immunity state, immunological disease, immune dysfunction) about the cause of the different immunogenicty and durability of antibody responses to homologous and heterologous vaccinations."

Author Response

* Reviewer 2

The article entitled “Immunogenicity and durability responses to homologous and hererologous vaccinations with BNT162b2 and ChAdOx1 vaccines for COVID-19” reports a study enrroling 69 participants vaccinated with COVId-19 vaccines, including BNT-BNT mRNA homologous vaccination and ChAd-ChAd viral-vector homologous vaccination and ChAd-BNT heretorologous vaccination to evaluate the immunogenicity and longevity of the antobody response to the original vaccine antigen. The authors revealed that the BNT-BNT group and ChAd-ChAd group showed strong induction and rapid decay of antibody response, and moderate induction and slow waning of antibodies, respectively. The ChAd-BNT group showed a potent antibody response, and the response remained at the highest level at approximately  6 months after the primary vaccination. The authors did not assess vaccine-induced T-cell and B-cell responses that in my opinion can provide a better understanding about the different immunogenicity and durability of vaccine-induced immune response.

Although there are many articles about this argument in the literature, I think that this manuscript reinforces the concept of the usefulness of vaccination practice and this is a strenght point in this manuscript. The weakness is that the authors would provide more clinical and biological details (immunity state, immunological disease, immune dysfunction) about the cause of the different immunogenicty and durability of antibody responses to homologous and heterologous vaccinations.

* Response: The authors appreciate the valuable opinion from the reviewer.

Regarding the assessment of T and B cell responses triggered by vaccines, we could not analyze B and T cell responses in this study due to insufficiency of PBMC specimens. These data could have explained many aspects of clinical details of these vaccinated individuals.

By the way, the authors could add a data describing previous SARS-CoV-2 infection history in Figure 1B by measuring anti-SARS-CoV-2 nucleocapsid antibodies.

Reviewer 3 Report

Here, Kim et al. tested the immunogenicity and durability of different vaccine regiments using spike specific IgG as the biomarker from sera sample of a total of 69 vaccinated individuals. These people have been prime-boost immunized either by homologous (BNT162b2-BNT162b2 or ChAdOx1-ChAdOx1) or heterologous (ChAdOx1-BNT162b2) vaccination. The end point is six months after the initial prime. They found better induction and more rapid decline of antibody responses in the cohort received homologous vaccination of the mRNA vaccine. On the contrary, they observed weaker boost and more stable antibody responses in the cohort received viral vector vaccine.

This is a well-organized study. It touched on a very important question facing in the field, the longevity of immunity triggered by vaccination against SARS2 viruses. The results will be interested to a broad audience and valuable for public officials to design a vaccine strategy to get us better prepared for the COVID-19.

Main Comment:

1.     The interval between primer and boost is different between mRNA vaccines and viral vector vaccines. This might play a role in the difference in immunity that they observed.

2.     Given the emergence of variant of concern (VOC), it would be more valuable to test the potency of those existed antibodies against VOC strains. Particularly for the sera sample collected at 6 months post vaccination.

Author Response

* Reviewer 3

Here, Kim et al. tested the immunogenicity and durability of different vaccine regiments using spike specific IgG as the biomarker from sera sample of a total of 69 vaccinated individuals. These people have been prime-boost immunized either by homologous (BNT162b2-BNT162b2 or ChAdOx1-ChAdOx1) or heterologous (ChAdOx1-BNT162b2) vaccination. The end point is six months after the initial prime. They found better induction and more rapid decline of antibody responses in the cohort received homologous vaccination of the mRNA vaccine. On the contrary, they observed weaker boost and more stable antibody responses in the cohort received viral vector vaccine.

This is a well-organized study. It touched on a very important question facing in the field, the longevity of immunity triggered by vaccination against SARS2 viruses. The results will be interested to a broad audience and valuable for public officials to design a vaccine strategy to get us better prepared for the COVID-19.

Main Comment:

  1. The interval between primer and boost is different between mRNA vaccines and viral vector vaccines. This might play a role in the difference in immunity that they observed.

* Response: The authors appreciate the important comment from the reviewer. Different intervals of vaccination would definitely change the outcomes of vaccine-induced immunity. We tried to extract valuable information as much as possible from the data by comparing ChAd-ChAd and ChAd-BNT groups since these groups have similar interval time.

In addition, as pointed by the reviewer, regarding the comparison between BNT-BNT and ChAd-BNT or ChAd-ChAd groups, it was not feasible to directly compare the kinetics of antibody responses due to the difference in intervals. However, the authors think that it would be valuable to roughly describe the overall changes of humoral responses in three vaccinated groups over 6 months.

  1. Given the emergence of variant of concern (VOC), it would be more valuable to test the potency of those existed antibodies against VOC strains. Particularly for the sera sample collected at 6 months post vaccination.

* Response: The authors agree with the valuable comment from the reviewer regarding the assessment of vaccine response to different VOCs, and that would have greatly enhanced the value of this manuscript. However, we though that many other studies reported about the vaccine efficacy to VOCs. Therefore, this study focused on the immunogenicity and durability of different regimen of COVID-19 vaccination by measuring humoral immune response to the parental strain of SARS-CoV-2.

Round 2

Reviewer 1 Report

The authors did a good job addressing my initial concerns. I only have one remaining comment: authors should provide a description of the anti-nucleocapsid IgG ELISA assay in the material & methods section.

Author Response

As pointed by the reviewer, we added technical information about the assessment of anti-nucleocapsid antibodies.

The authors really appreciate the reviewer for taking time for review, kind comments and the favorable decision. 

Reviewer 2 Report

I read the revised version of the article entitled “ Immnunigenicity and durability of antibody responses to homologous and heterologous vaccinations with BNT162b2 and ChAdOx1 vaccines for COVID-19”. Although the authors have not added the findings regarding the T and B cell responses in the vaccinated patients I think that the mesurement of the antinucleocapsid IgG can be a marker of T and B cell immunity. Therefore, I think that the revised version of this article is suitable for publication.

Author Response

The authors really appreciate the reviewer for taking time for review, kind comments and the favorable decision.